# Density of CD3+ and CD8+ cells in gingivo-buccal oral squamous cell carcinoma is associated with lymph node metastases and survival

Geetashree Mukherjee[1]☯*, Swarnendu Bag[1]☯, Prasenjit Chakraborty[1], Debdeep Dey[1], Samrat Roy[1], Prateek Jain[1], Paromita Roy[1], Richie Soong[2], Partha Pratim Majumder[3], Suparna Dutt[4]*

**1** Tata Medical Centre, Kolkata, India, **2** Pacific Laboratories, Singapore, Singapore, **3** National Institute of Biomedical Genomics, Kalyani, West Bengal, India, **4** Division of Immunology and Rheumatology, Department of Medicine, Stanford University School of Medicine, Stanford, California, United States of America

☯ These authors contributed equally to this work.
* sdutt@stanford.edu (SD); mukherjeegeeta@yahoo.co.in (GM)

**Data Availability Statement:** The patient data is available from the SyMeC Data Centre: http://symec.isical.ac.in/symec/Oral_GB_TMC_IML.php.

## Abstract

The tumor immune microenvironment is emerging as a critical player in predicting cancer prognosis and response to therapies. However, the prognostic value of tumor-infiltrating immune cells in Gingivo-Buccal Oral Squamous Cell Carcinoma (GBOSCC) and their association with tumor size or lymph node metastases status require further elucidation. To study the relationship of tumor-infiltrating immune cells with tumor size (T stage) and lymph node metastases (N stages), we analyzed the density of tumor-infiltrating immune cells in archived, whole tumor resections from 94 patients. We characterized these sections by immune-histochemistry using 12 markers and enumerated tumor-infiltrating immune cells at the invasive margins (IM) and centers of tumors (CT). We observed that a higher density of CD3+ cells in the IM and CT was associated with smaller tumor size (T1-T2 stage). Fewer CD3+ cells was associated with larger tumor size (T3-T4 stage). High infiltration of CD3+ and CD8+ cells in IM and CT as well as high CD4+ cell infiltrates in the IM was significantly associated with the absence of lymph node metastases. High infiltrates of CD3+ and CD8+ cells in CT was associated with significantly improved survival. Our results illustrate that the densities and spatial distribution of CD3+ and CD8+ cell infiltrates in primary GBOSCC tumors is predictive of disease progression and survival. Based on our findings, we recommend incorporating immune cell quantification in the TNM classification and routine histopathology reporting of GBOSCC. Immune cell quantification in CT and IM may help predict the efficacy of future therapies.

**Funding:** This study was funded by Department of Biotechnology, Government of India, for funding the project under Systems Medicine Cluster (Project name: "Multi-dimensional Research to Enable Systems Medicine: Acceleration Using a Cluster Approach". Reference No: BT/Med-II/NIBMG/SyMeC/2014/Vol. II, dated 09/01/2017) to Dr. Partha Pratim Majumder. The funders had no role in study design, data collection and analysis, decision to publish, or preparation of the manuscript. Dr. Prasenjit Chakraborty and Dr. Swarnendu Bag received salary support from the above-mentioned funding.

**Competing interests:** NO authors have competing interests.

# Introduction

Chewing tobacco is a habit very prevalent in India. It is the strongest risk factor for the development of oral cancer. Oral cancer comprises about 12% of all male cancers in India, of which about 40% are gingivobuccal [1, 2].The incidence of Oral Squamous Cell Carcinoma of the Gingivo-Buccal region (GBOSCC) includes buccal mucosa, gingivo-buccal sulcus, alveolus and retro-molar trigone. India has one of the highest incidences of this form of cancer in the world. Despite advances made in treatment modalities, locoregional recurrence is the primary cause of treatment failure in advanced stages of the disease [3] with a dismal 5-year survival rate between 5–15% [4]. Nodal metastases is the most significant adverse prognostic factor of GBOSCC survival [5].

In recent years, advances in immunotherapy have had a major impact on cancer treatment. The effectiveness of immunotherapy for a patient depends largely on the presence of a baseline tumor immune profile [6–10]. The composition of tumor immune microenvironment in oral squamous cell carcinoma not only influences the disease pathogenesis [11–13] but also is a strong prognostic indicator of clinical response to treatments [14].

The current classification consists of three types of tumor immune microenvironments, (TME) namely immune hot, immune cold, and immune altered [15–19]. This distinction is based on the distribution of cytotoxic CD8$^+$ T cells in the tumor microenvironment (TME) and has been described in cancers such as melanoma [20] and colorectal cancer [21]. Hot immune tumors have high infiltration of CD3$^+$ and CD8$^+$ T cells in the invasive margin (IM) and center of tumor (CT), while in cold tumors, there is the absence of T cell infiltrates within the IM and CT. Immune altered tumors are characterized by the accumulation of T cells at the IM only (altered excluded) or minimal infiltration of T cells within the CT (altered immuno-suppressed) [15]. Studies have linked spatial organization of immune cells in the TME to the clinical outcome by gene expression profiling and immuno-histochemistry (IHC) [22–26]. Prognostic and predictive signatures were derived from gene expression and IHC profiles [9]. These signatures indicated the complex interplay between the TME and the immune system [15]. The clinical significance of tumor infiltrating immune cell density and their spatial location in TME has led to the development of immunoscore-a cytotoxic immune signature. This was first described by Galon et al. in colorectal cancer [19, 21], where the density of T cells was measured as a "score" both at the invasive tumor margin and centers of tumor. A consensus immunoscore, categorizing inflamed and non-inflamed tumors, was subsequently validated with high clinical relevance in colorectal cancer [27]. A strong correlation was observed between several previously published transcriptional signatures reflective of the T cell inflamed TME and cytolytic processes from The Cancer Genome Atlas (TCGA) data set [28].

The TME also drives metastasis along with factors intrinsic to tumors [29]. Tumor metastasis is promoted, at least partially, by interactions between tumor and immune cells through the secretion of cytokines, growth factors and proteases that remodel the TME [29]. Communication between cancer cells with various stromal cells of the TME also promotes metastasis [30, 31]. In colorectal cancer, reduced immune cytoxicity (low immunoscore) was more strongly associated with distant metastases than tumor intrinsic factors such as chromosomal instability and mutation burden [32]. Absence of regional lymph node metastasis in oral cancers was associated with high numbers of CD8$^+$ cells in tumors [33]. Lymph node metastasis and tumor size have been found to be strongly prognostic of survival in GBOSCC [34]. Given an alarmingly high incidence of GB OSCC cases in India, there is an urgent need to develop a tool to predict which patients may develop lymph node metastasis and may have poor clinical outcomes. We hypothesized that the baseline density of tumor infiltrating immune cells and their spatial distribution in distinct regions of primary GBOSCC are critical determinants of lymph

node metastasis and overall patient survival. Here, we utilized immuno-histochemical staining of surgically resected, whole tumor, histopathology sections to quantify the density of tumor-infiltrating cells in the IM and CT to predict lymph node metastasis and survival.

## Materials and methods

### Patient tumor tissues

All GBOSCC patients who underwent surgery with lymph node dissection at the Tata Medical Centre (TMC), Kolkata, India, between 2012 to 2014 were selected for this study. None of the patients received prior treatment at TMC. Patients who received prior treatment elsewhere were excluded. Patients were followed up every 2 months for the initial 2 years. The follow-up interval was then increased gradually. Each patient was re-contacted and relevant data such as complaints, general conditions, symptoms and history of recurrence were collected. If there was any complaint that patients had, they were advised to immediately visit the hospital, or get medical advice from a hospital near their home. The data was collected for at least two years by telephone calls or during their follow-up visits to the hospital. The median follow-up time was 50 months (range: 29–93 months). Overall survival (OS) was determined based on the date of diagnosis until the date of death or the date of last follow up at the end of study. Clinical and pathology details of each patient were collected from hospital records. All patient data were fully anonymized before being accessed. Grading and staging were performed according to the WHO classification of tumors and UICC TNM classification at the time of diagnosis [35]. The presence of tumor cells in the lymph nodes or node status (N status) was determined patholog-ically. All patients received standard-of-care treatment according to National Comprehensive Cancer Network (NCCN) guidelines [36, 37]. Archived formalin-fixed paraffin-embedded (FFPE) paraffin blocks of resected tumor specimens were retrieved and reviewed for each patient, and only block containing at least 50% tumor tissue, were selected for analysis. This study was approved by the Institutional Review Board of Tata Medical Center (IRB No: EC/TMC/69/16). As this was a retrospective study of anonymized samples, consents from patients was not obtained. A waiver of consent was given by the Institutional Review Board.

### Single-marker immuno-histochemistry (IHC)

From each block, 3μm-thick sections were prepared and dried in a 60˚C oven overnight. IHC staining of the sections was performed in a Bond Max Automated Immuno-histochemistry Vision Bio-system (Leica Microsystems GmbH, Wetzlar, Germany) according to standardized protocols. First, tissues were de-paraffinized and pre-treated with the Epitope Retrieval Solu-tion 2 (pH8.9–9.1) at 98˚C for 20 min. After washing with wash buffer, peroxidase blocking was carried out for 10 min using the Bond Polymer Refine Detection Kit DC9800 (Leica). Tis-sues were again washed, and then incubated with primary antibody for 30 min. We selected immune markers related to the adaptive and innate immunity. These included antibodies for CD3 (T cell receptor), CD4 (a CD3 co-receptor expressed on helper T cells), CD8 (a CD3 co-receptor expressed on cytotoxic T cells), Granzyme B (a serine protease expressed by CD8 T cells), CD68 (expressed in cytoplasmic granules of monocytes, macrophages, dendritic cells and granulocytes), neutrophil elastase (a serine protease found in polymorphonuclear neutro-phils), HLA-DR (a human leukocyte antigen class II molecule involved in antigen presentation to CD4 T cells expressed on macrophages, dendritic cells, monocytes, activated T cells and B cells), CD15 (glycoprotein expressed on polymorphonuclear granulocytes), CD14 (expressed on macrophages, monocytes, and neutrophils), Arginase1 (a metalloenzyme that catabolizes arginine expressed by myeloid derived suppressor cells and is involved in T cell suppression), CD56 (expressed on NK cells) and CD20 (expressed on B cells). Following incubation with

primary antibodies, tissue sections were incubated with polymer for 10 min and developed with DAB-Chromogen for 10 min. After counterstaining with hematoxylin (Dako, Jena, Germany), slides were dehydrated and mounted with mounting medium (Dako). To test the specificity of the staining protocols, sections were also stained without primary antibody. No non-specific staining in such sections was observed. Lymph node sections were included in each staining batch as a positive control. Consecutive slides were used for the 12-marker single IHC staining. Antibodies: CD3 (Dako; Rabbit polyclonal,1:200), CD8 (Dako; clone C8/144B, 1:50), CD4 (PathinSitu; EP204, RTU), CD68 (Dako; PGM1, RTU), Granzyme B (Dako; clone GrB-7, 1:25), Neutrophil Elastase (Dako; clone NP57, 1:100 dilution), CD15 (Dako; clone carb-3, 1:200), HLA-DR (PathnSitu; clone EP-128, RTU), Arginase1 (Abcam; clone EPR6672, 1:250), CD14 (Master Diagnostics; clone EP128, RTU), CD20 (Dako; clone L26, 1:500), CD56 (PathnSitu; clone 123C3, RTU). For Human Papillomavirus (HPV) testing, p16 staining was performed in sections using primary antibody (E6H4) mouse monoclonal antibody (RTU, Roche) on the Ventana platform. A positive test was based on 8th Edition of UICC TNM classification [38].

## Slide image analysis

Digital images of the stained slides were captured using the MANTRA slide imaging system (Perkin Elmer, Marlborough, MA) at 10x and 20x magnification. For every slide, each marker was measured in 2 spatial compartments of the tissue sections, namely 1) invasive margin of tumor defined as 1mm of the invasive edge at the interface of tumor and normal tissue 2) the center of tumor, defined as any other region within the tumor except the invasive margin [39, 40]. Five different regions from invasive margin and 5 regions from tumor center were selected by the pathologist in order to cover the entire tumor and the average scores were taken for each of the sites. Only images captured at 20x magnification were analyzed using the InForm 2.4 software (Perkin-Elmer). Immune cell levels were quantified as the percentage of cells with a minimal intensity that was considered positive, by two pathologists [41]. In a small number of cases where scoring seemed to be inaccurate due to mild background staining (5%), the slides were checked manually by two pathologists and the average score of the independent assessment of the pathologists was considered.

## Statistical analysis

The distribution pattern for all immune marker' expression was checked by Shapiro-Wilk normality test [$p < 0.05$ (CI 95)]. Mann-Whitney U test was performed to compare the expression level of immune cell marker between node-positive vs negative groups and tumor size T1-T2 vs T3-T4 groups. MANOVA was used for Multivariate analysis. This analysis included the expression of immune cell markers that were significant between node-positive vs negative groups according to Mann-Whitney U test along with clinicopathological variables Lympho-vascular invasion (LVI), Perineural invasion (PNI) and tumor size. Two Onco-pathologists provided Immune-marker-expression-score for each subject. The scoring system was as follows: Low– 0–25%; Intermediate- 26–50%, High- 51–100%. Mann-Whitney U test was performed to evaluate whether the scoring system adopted here can significantly classify patients based on node status and tumor size,. K-Nearest Neighbors (KNN) based supervised machine learning algorithm was used to predict any plausible classification between the expression of immune cell markers and node positivity. Kaplan-Meier survival analysis was performed to determine the association of immune marker expression with survival. Statistical analysis was carried out using R (R-3.6.2) and SPSS (IBM SPSS Statistics 23).$P < 0.05$ was considered statistically significant for all analyses.

# Results

## Patient characteristics

Clinical and Pathological characteristics of 94 patients are listed in Table 1. Of the 42 node-negative patients, 3 had local recurrence during follow up, but none developed lymph node metastasis during follow up. Patients who had distant metastasis were all node-positive at diagnosis. Human papilloma virus (HPV) infection is an established risk factor, with prevalence in oral squamous cell carcinoma [42], we performed p16 staining for HPV in all 94 cases. One showed p16 positive staining with >70% diffuse and strong nuclear and cytoplasmic staining. Three cases showed equivocal staining with <70% but >50% diffuse and strong nuclear and cytoplasmic staining and rest 90 cases were negative (<50% diffuse and strong nuclear and cytoplasmic staining) (S1 Fig).

## Association of immune cell infiltrates at the invasive margin and center of tumors with tumor size and lymph node metastasis

All tumors in our study belonged to either hot tumor or immune altered (immune excluded or immunosuppressed) types [15]. We stained for 12 immune cell markers of the innate and adaptive immunity (S1 Fig). The association of each immune cell density with tumor size (T stage) and lymph node metastasis status (N stage) of the disease were determined for each surgically resected tumor section. Fig 1A shows H&E staining of a representative resected whole tumor. Fig 1B and 1C show CD3 and CD8 staining at the IM and CT respectively. S1 Table. shows the results of Shapiro-Wilk normality test [P<0.05 (95% CI)] for distribution pattern of expression of immune cell markers. The presence of a high percentage of CD3+cells both in IM (p = 0.006 95% CI) and CT (p = 0.003 95% CI) was associated with small tumor size (T1 & T2). No statistically significant difference (P>0.05) in the expression CD15, CD14, and CD56 were observed between the T1&T2 and T3&T4 groups (S2 Fig). However, there were significant differences in the expression of Granzyme B between T1-T2 vs T3-T4 in both IM

**Table 1. Patient and tumor characteristics.**

| Characteristics | Data (n = 94) |
|---|---|
| Age | 30–85 years (Median-55.5 years) |
| Gender | Male 70 (74.4%) |
| | Female 24(25.53%) |
| Tumor Size (T1&T2) | 41 (43.62%) |
| Tumor Size (T3&T4) | 53 (56.38%) |
| Node Negative | 42(44.68%) |
| Node Positive | 52(55.32%) |
| Tumor grade | I– 3 |
| | II– 88 (93%) |
| | III– 3 |
| Lympho-vascular invasion | Present– 50 (53%); |
| | Absent– 44(47%) |
| Mucosal margins | Positive-2 |
| | Very close-2 |
| | Negative-90 (96%) |
| Perineural invasion | Present-41(44%) |
| | Absent-53(56%) |
| Distant metastasis | M0 at diagnosis. (5 patients developed distant metastasis within one year of diagnosis). |

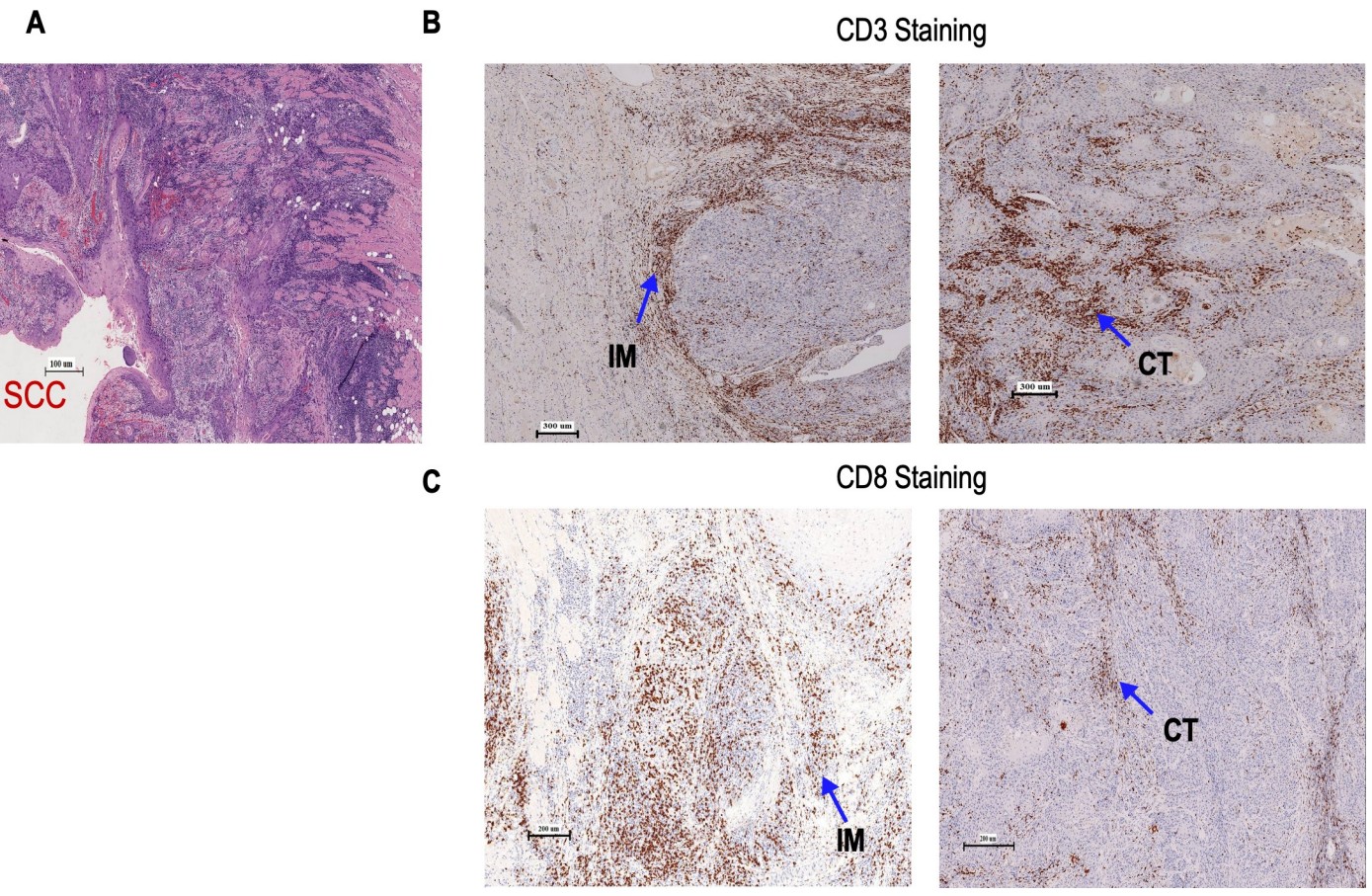

**Fig 1. Location of Immune infiltrates in GBOSCC.** (A) Representative H & E staining (200X) of GBOSCC; (B) Shows staining of CD3$^+$ cells (20X) at the IM and CT; (C) Shows staining of CD8$^+$ cells (20X) at the IM and CT.

(P = 0.003 95%CI) and CT (P = 0.018 95% CI). We also observed significant differences in expression of Neutrophil Elastase and Arginase1 (P = 0.003 95%CI) in IM between T1-T2 vs T3-T4 (P = 0.047 95%CI). (S2 Fig). An increase in CD68$^+$ cells in the CT (p = 0.015 95% CI) was associated with smaller tumor size (T1&T2) (Fig 2). Fig 3 shows the mean percentages of CD3 $^+$, CD8$^+$, and CD68 $^+$ cells–both in the IM and CT that were significantly different between patients with (N+) and without (N0) lymph node metastasis. Patients with high percentages of CD3$^+$, CD8$^+$, and CD68$^+$ cells at the IM and CT, had fewer lymph node metastasis. (CD3–CT p = 0.010 95% CI, CD3-IM p≤0.001; CD8–CT p = 0.001 95% CI, CD8-IM p = 0.002 95% CI; CD68–CT p≤0.001 95% CI, CD68-IM p≤0.001 95% CI). A high count of CD4$^+$ cells at the IM was associated with node negativity (CD4–IM p = 0.005) whereas, high HLA-DR expression at the IM was associated with lymph node metastasis. There were no significant differences in the expression of Granzyme B, neutrophil elastase, CD15, CD14, Arginase1, CD56 between the lymph node metastasis-positive and -negative groups (S3 Fig).

## Association of immune marker expression score with tumor size and lymph node metastasis

The association of immune cell marker expression at IM and CT with lymph node metastasis was further evaluated through 'Line of Best Fit' method. Scatter plot represents the coordinate

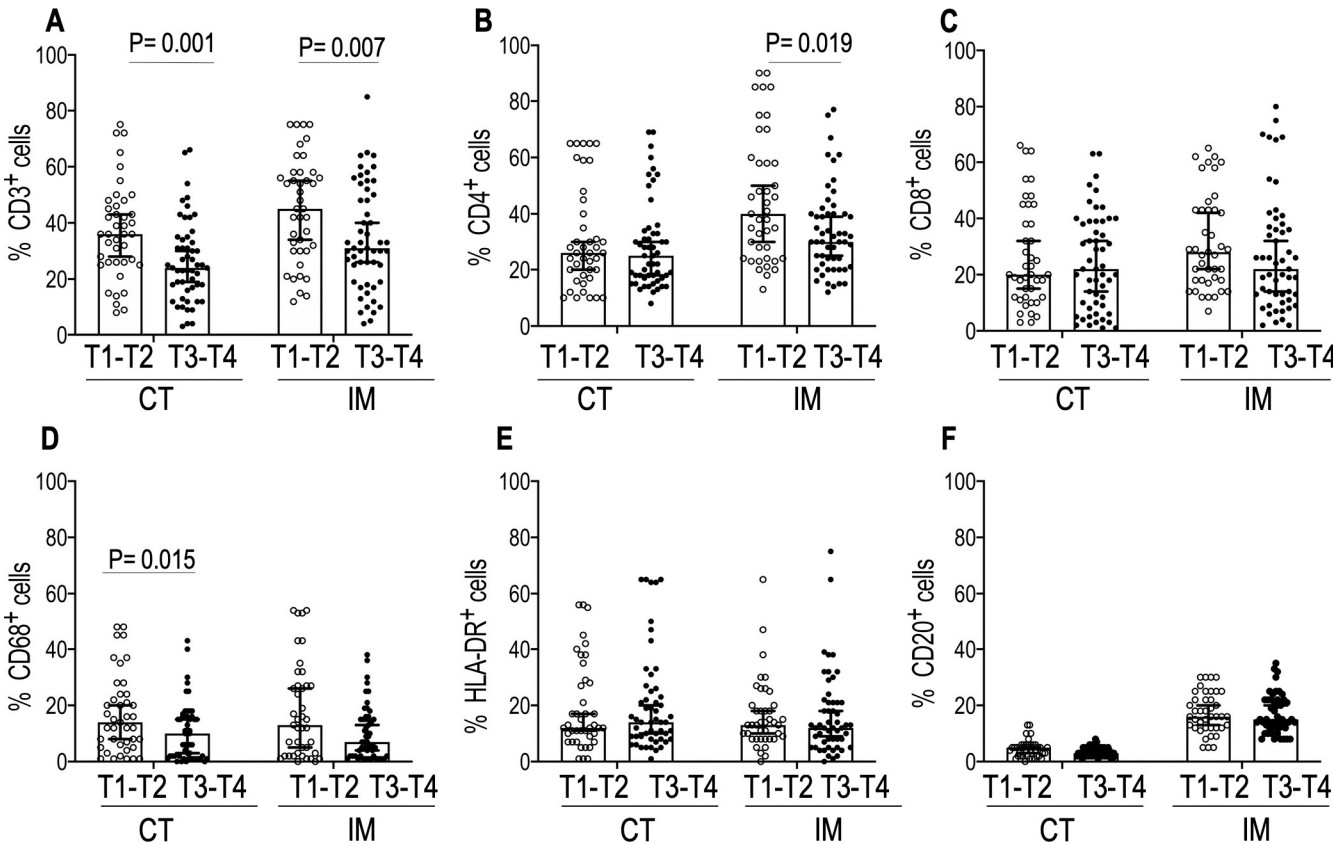

**Fig 2. Comparisons of the densities of immune cells in IM and CT between Group T1- T2 and Group 2 T3-T4.** PerkinElmer inForm software was used to enumerate densities of immune cells. Data represented are median with 95% CI. Two-tailed Mann -Whitney U test was performed to test statistical significance. Open circle denotes T1-T2 and the closed circle denotes T3-T4. T1-T2 (n = 41) and T3-T4(53). P≤ 0.05 was considered significant. Significant P values are shown.

position of each marker expression (mean value) (Fig 4A). The Line of Best Fit shows that the expressions of CD3 and CD8 in both IM and CT regions significantly (p<0.05) discriminate the lymph node metastasis positive and negative patients (Fig 4A). This finding prompted us to further analyze the association of CD3 and CD8 with node status utilizing immune marker expression score analyses. The percentages of CD3$^+$ and CD8$^+$ cells in CT [Low = 0–25%; Intermediate = 26–50%; High = 51–100%] were given independently by two Onco-Pathologists. Fig 4B showed that patients with node negative status had significantly higher (p<0.05) 'High' and 'Intermediate' immune marker expression scores in CT compared to node positive patients. We further analyzed the immune marker expression score to classify patients according to immune infiltrate and tumor size. Smaller tumors (T1-T2 stage) were associated with higher CD3$^+$ and CD8$^+$ immune marker expression score at the CT. A decrease in CD3$^+$ and CD8$^+$ cells in CT was associated with lymph node metastasis. Furthermore, classification matrix (Fig 4C) derived from supervised machine learning based KNN classification model performed in R (R-3.6.2) showed that CD8$^+$ cells in CT had strong association with node status and tumor size with accuracy level of 72.34% and sensitivity of 94.23% for node status and accuracy level of 73.40% & sensitivity of 91.07% for tumor size. [For Node status: TN (True Negative): 19; TP (True Positive) = 49; FP (False Positive) = 23; FN (False Negative) = 3; Sensitivity = {TP/(TP+FN)}x100 = {49/(49+3)}x100 = 94.23% & Accuracy = {(TP+TN)/(TP+FP +TN+FN)}x100 = {(49+19)/(49+23+19+3)}x100 = 72.34%; For Tumor size: TN (True

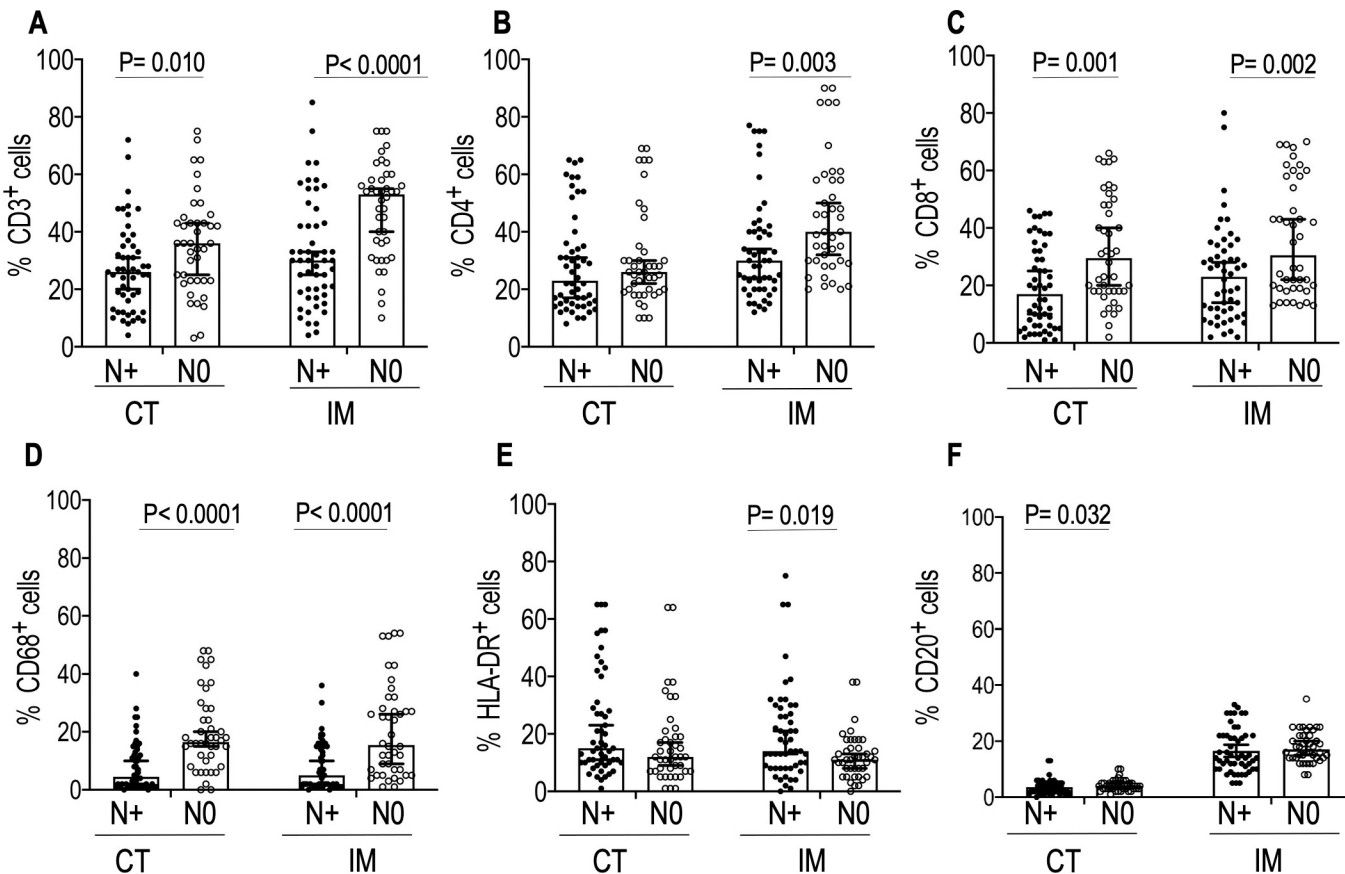

**Fig 3. Comparisons of the densities of immune markers in IM and CT in primary tumors between lymph node positive (N+) and lymph node negative (N0) patients.** Perkin Elmer InForm software was used to enumerate densities of immune cells. Data represented are median with 95% CI. Two-tailed Mann-Whitney U test was performed to test statistical significance. Closed circle denotes N+ and the open circle denotes N0. N+ (n = 52) and N0(42). Significant P values are shown.

Negative): 18; TP (True Positive) = 51; FP (False Positive) = 20; FN (False Negative) = 5; Sensitivity = {TP/(TP+FN)}x100 = {51/(51+5)}x100 = 91.07% & Accuracy = {(TP+TN)/(TP+FP+TN+FN)}x100 = {(51+18)/(51+20+18+5)}x100 = 73.40%]

## Multivariate analyses of immune markers and pathological parameters to node status

Table 2 shows multivariate analyses of percent expression of immune cell markers with Lympho-Vascular Invasion (LVI), Perineural Invasion (PNI) and tumor size to node status. All variables except tumor size were found to significantly discriminate between node-negative and node-positive patients.

## Densities of CD3+ and CD8+ cells at the center of tumor correlate with survival in GBOSCC patients

To determine whether the location and densities of immune cells in tumors contribute to overall survival, we performed Kaplan-Meier survival analysis (Fig 5). Considerable variation in the number of days of survival after treatment was observed. We found significant associations between overall survival and high densities of CD3+as well as CD8+ cells (P< 0.05) at the CT

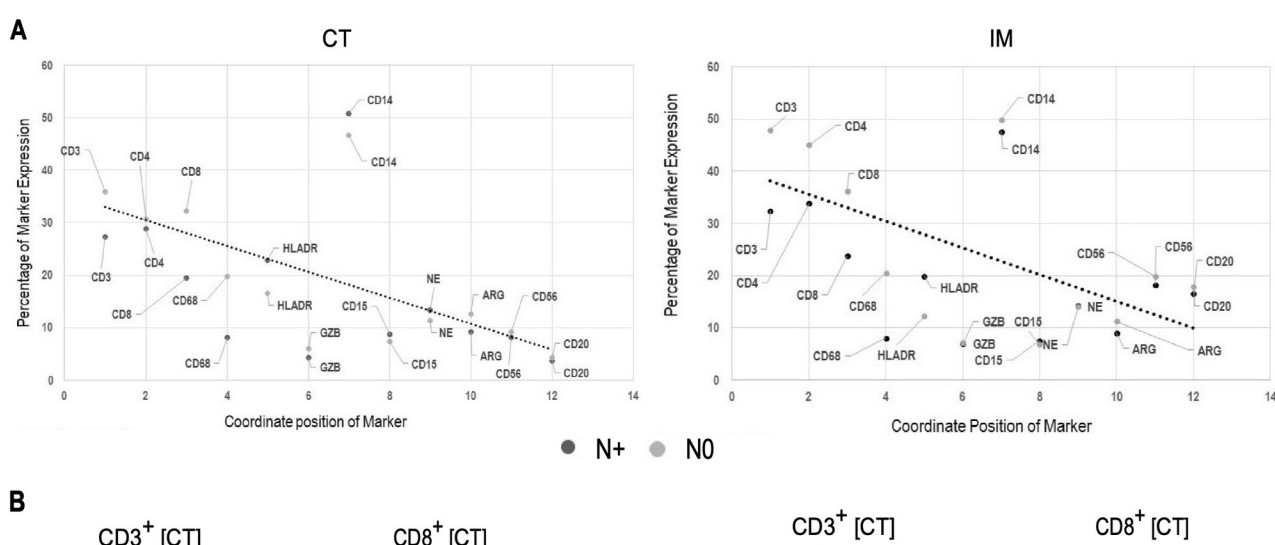

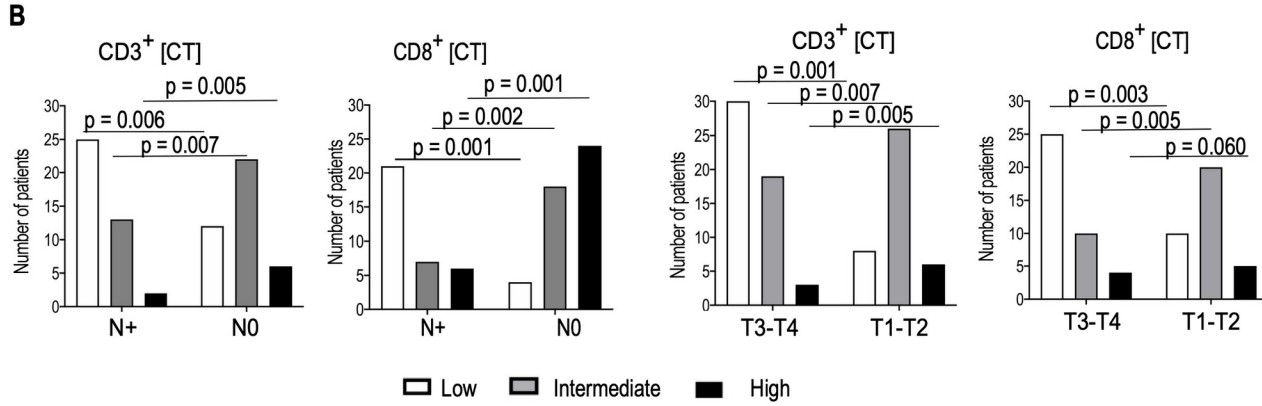

**C**

T stage

| N= 94 | Predicted : No | Predicted :Yes | Row Total |
|---|---|---|---|
| Actual :No | 18(47.37%) | 20(52.63%) | 38 |
| Actual :Yes | 5(8.93%) | 51(91.07%) | 56 |
| Column Total | 23(24.47%) | 71(75.53%) | 94 |

N stage

| N= 94 | Predicted : No | Predicted :Yes | Row Total |
|---|---|---|---|
| Actual :No | 19(45.24%) | 23(54.76%) | 42 |
| Actual :Yes | 3(5.77%) | 49(94.23%) | 52 |
| Column Total | 22(23.40%) | 72(76.60%) | 94 |

**Fig 4. Association of CD3+ and CD8+ cells with tumor size and lymph node metastasis and comparison of expression of immune marker score at the IM and CT in primary tumors between patients in the T1-T2 and T3-T4 groups and lymph node positive (N+) and lymph node negative (N0) patients.** (A) represents the line of best fit amongst each marker expression (mean value); (B) depicts the distribution of immune cell marker expression score(High, Intermediate, Low) between T1—T2 and T3—T4 groups (left) and node- positive (N+) and node-negative (N0) patients(right). (C) Classification matrix derived from KNN machine learning model to predict T1-T2 and T3-T4 status based on CD8 expression in CT with Accuracy: 73.40 & Specificity: 91.07 and N+ and N0 status, based on CD8 expression in CT with Accuracy: 72.34% & Specificity: 94.23%.

**Table 2. Multivariate analyses of the following variables with respect to node status of GB-OSCC patients.**

| Multivariate Tests | | | | | |
|---|---|---|---|---|---|
| Effect | | Value | F | Hypothesis df | Error df | Sig. |
| NODE | Pillai's Trace | .463 | 9.149 | 8.000 | 85.000 | .000 |
| | Wilks' Lambda | .537 | 9.149 | 8.000 | 85.000 | .000 |
| | Hotelling's Trace | .861 | 9.149 | 8.000 | 85.000 | .000 |
| | Roy's Largest Root | .861 | 9.149 | 8.000 | 85.000 | .000 |
| **Tests of Between-Subjects Effects** | | | | | |
| Source | Dependent Variable | Type III Sum of Squares | df | Mean Square | F | Sig. |
| NODE | PNI | 2.306 | 1 | 2.306 | 10.193 | .002 |
| | LVI | 8.694 | 1 | 8.694 | 54.653 | .000 |
| | T | 2.979 | 1 | 2.979 | 2.199 | .142 |
| | CD3CT | 1738.503 | 1 | 1738.503 | 6.855 | .010 |
| | CD3IM | 5537.875 | 1 | 5537.875 | 17.535 | .000 |
| | CD4IM | 2910.476 | 1 | 2910.476 | 8.122 | .005 |
| | CD8CT | 3841.277 | 1 | 3841.277 | 14.294 | .000 |
| | CD8IM | 3593.181 | 1 | 3593.181 | 10.840 | .001 |

(Fig 5A and 5B). In contrast, there were no significant association between high densities of these cells at the IM with survival (Fig 5C and 5D).

## Discussion

In the present retrospective study, we investigated the immune cell infiltration using 12 markers. High counts of CD3 $^+$ and CD68$^+$ cells at the IM and CT were associated with smaller tumor size. Smaller tumors (T1-T2) had higher percentage of CD3$^+$, CD8$^+$ and CD68$^+$ cells compared to larger tumors (T3-T4). There is evidence that presence of high numbers of CD3$^+$ T cells are associated with good prognosis and increased survival in oral squamous cell carcinoma [43–45]. However, there are no studies where CD3$^+$ cell count has been correlated with tumor size. One study showed that patients with leukoplakia without malignant transformation had higher numbers of CD3$^+$ cells than patients whose leukoplakia transformed to oral squamous cell carcinoma [46]. Our results agree with existing reports [47–49] that presence of a baseline anti-tumor immune cell infiltrate or pre-existing tumor immunity [21] is associated with high levels of tumor infiltrating CD8$^+$ T cells and strongly associated with improved patient survival. Granzyme B$^+$ cells were found to be significantly higher in IM and CT of T1-T2 stage tumors compared to T3-T4 tumors suggesting that small tumor size may be attributed to Granzyme B mediated killing of tumor cells [50]. Our results also showed that smaller tumors (T1-T2) tumors were associated with significant increase in Neutrophil Elastase$^+$ and Arginase I$^+$ cells at the IM than T3-T4 stage tumors. Tumor associated neutrophils release Arginase-I during activation that suppress T cell responses [51]. Therefore, the increase in Arginase I$^+$ cells in IM of T1-T2 tumors could be related to increase in activated neutrophils. Si et al showed that Granzyme B expression was reduced in T cells in the proximity of immunosuppressive neutrophils in head neck cancer tissues [52]. Further analysis is required to characterize the phenotype of the neutrophils and determine their spatial distribution and interaction with T cells in our tumor tissue samples.

The most striking results in our present study were high CD3 and CD8 densities both in the IM and CT. We also found high counts of CD4$^+$ cells at the IM which were associated with significantly smaller tumor size (T1-T2 stage) and lymph node-negative disease at diagnosis. High levels of tumor infiltrating activated CD4$^+$ CD69$^+$ T cells are associated with better

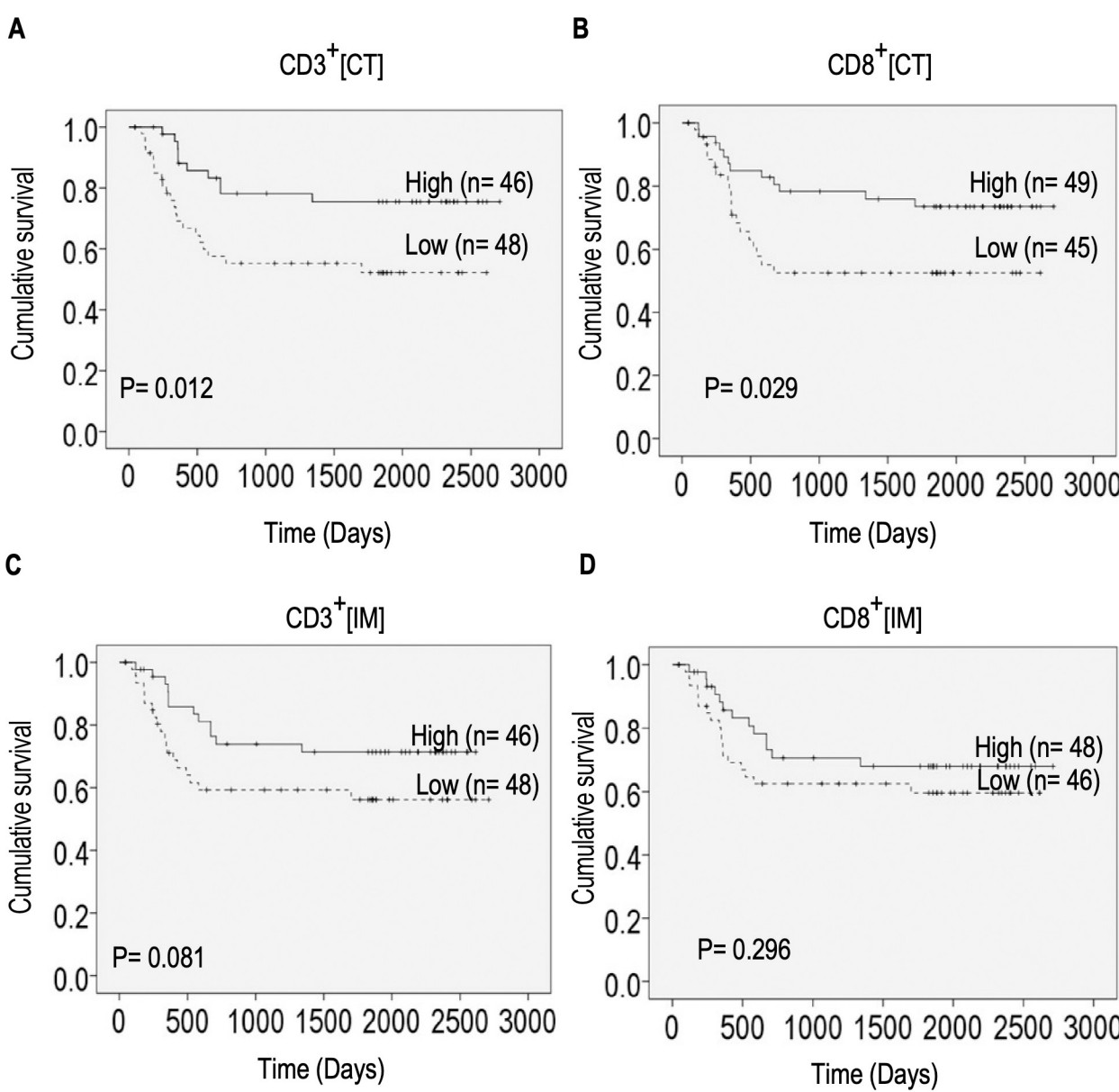

**Fig 5. Effect of density of immune cell infiltrates at the IM and CT on survival. Kaplan-Meier curves comparing overall survival in patients with high and low densities of CD3+ cells in the CT.** (A), IM (C) and high and low densities of CD8+ cells in the tumor CT (B), IM (D). Mean cell percentages were used as threshold to stratify patients in low and high groups. The cutoff threshold at CT were 31.09% (CD3+) and 25.15% (CD8+), and at IM were 39.13%(CD3+) and 29.28% (CD8+). P values between groups were calculated by log-rank test.

locoregional control and improved survival in Head and Neck Squamous Cell Carcinoma patients [53]. The CD4+ cells in the lymph node negative or T1-T2 stage patients may be activated T cells that are known to promote activation cytolytic CD8+ T cells [54]. Our multivariate analysis showed significant association of Lympho-Vascular Invasion (LVI) and Perineural Invasion (PNI) with lymph node metastases. Similar results were reported in Oral Squamous Cell Carcinoma [55–57]. An established link between tumor infiltrating cells and lymph node metastasis has been reported in 78 cases of oral squamous cell carcinoma, where increased CD8+ cells was associated with absence of lymph node metastasis [33]. Another study used tissue microarrays (TMA) [58] to investigate the role of tumor infiltrating cells to predict patient

outcome. They found that CD68[+] macrophages were higher in patients with positive lymph nodes. We observed that expression of CD3[+] and CD8[+] cells in CT significantly correlated with patient survival. Improved survival may be likely due to effective anti-tumor immune responses as a result of the ability of cytotoxic T cells to infiltrate CT and establish physical contact with tumor cells and tumor killing. On the other hand, cytotoxic T cells at the IM may have the ability to mount anti-tumor response but tumor cells escape killing by hindering T cell infiltration into the tumor [15]. Similar results were obtained by Zhang et al. in head and neck cancer [59]. However, there were only 7 cases of oropharynx and oral cancer in this study. The authors used Immunoscore analysis to determine prognostic significance of TILs (tumor infiltrating cells) in resected tumors. Our study used immune marker expression score analysis for the first time in GBOSCC to determine the association of tumor infiltrating lymphocytes (TILs) with lymph node metastasis and patient survival. The majority of our patients without nodal involvement had baseline high or medium CD8[+] immune cell marker expression score at the center of tumor. High immunoscores based on quantification of densities of CD3[+] and CD8[+] in IM and CT in primary tumors has been shown to be associated with lower metastases in colon cancer patients [32, 60]. Our results suggest that high levels of cytotoxic CD8[+] T cell infiltration in GBOSCC tumors may be necessary to induce robust T cell anti-tumor response that may prevent tumor invasion and subsequent development of metastases. In our study, the patients with lymph node metastases had mostly low immune cell marker expression score and displayed immune altered pattern with either T cells located at the invasive margin (immune excluded) or low infiltrations at the tumor core (immunosuppressed). The lack of T cell infiltration in the tumor core could indicate immunological ignorance—inability of adaptive immunity to recognize tumors [61]. Low T cell infiltrates in immunosuppressed phenotype suggests that immunosuppressive tumor microenvironment limits infiltration and expansion of T cells. Further analyses of expression of T cell checkpoints such as PD-1, CTLA4, LAG3 and TIM-3 and FOXP3[+] regulatory T cells as well as chemokines involved in T cell trafficking [15] are necessary to determine the contribution of immune cells and soluble factors in shaping the immune altered pattern of TME in GBOSCC tumors.

In our study, high CD68 density had association with lymph node negative disease. This immune marker is expressed by macrophages and cells in the monocyte lineage and is the most commonly used macrophage marker in human tissue. it is a pan macrophage marker that does not discriminate between tumoricidal M1 macrophages and anti-inflammatory expressing M2 macrophages. A preliminary study in oral cavity carcinoma reported that lymph node metastasis was associated with high levels of CD68[+] macrophages in tumors [58]. Another study in OSCC reported that expression of CD163 significantly correlated with overall survival while expression of CD68 did not [62]. A meta-analysis study in OSCC showed that CD163 is a more reliable prognostic marker than CD68 [44]. In contrast to these studies, our results showed that high density of CD68[+] cells correlated with small tumor size. The CD68[+] cells in our study may be pro-inflammatory M1 macrophages. Therefore, staining of co-stimulatory molecule CD80 expressed by pro-inflammatory macrophages along with CD163 is necessary to characterize these macrophages. Interestingly, Pinto *et al* showed that CD80[+] cells were more abundant in intra-tumoral region and invasive front of less invasive T1 stage colorectal cancers [63]. We found high percentages of HLA-DR[+] cells at the tumor margin that correlated with node involvement. HLA-DR is expressed constitutively in antigen presenting cells, macrophages, B-cells and dendritic cells [64]. Binding of T cells with their T cell receptor to the respective HLA-DR molecule on the antigen presenting cells, can result in T cell activation. Therefore, HLA-DR by itself, is not a biomarker of significance in oral cancer.

Previous studies that have explored intra-tumoral cellular and immune diversity in head and neck squamous cell carcinomas (HNSCCs) have mainly focused on the presence or

absence of Human Papilloma Virus (HPV) in the tumor [65–67] and have included all sub-sites. Therefore, the immune biomarkers are likely to differ between different tumor subsites and tumor stages. There is one study that reported immune cell infiltrates in homogeneous subsites [68]. Besides, in most of the studies IHC were performed on tissue microarrays (TMAs) and used different evaluation criteria, with small sample size [33, 58, 59]. The concept of immunologically hot and cold tumors has not been examined in most studies. Most of our patients were HPV negative. This result is consistent with a previous GB-OSCC study in which only 4% of patients were found to be HPV positive [69]. Other studies have reported similar low frequencies of HPV positivity in OSCC patients [70–72]. Our study comprised a homogeneous group of patients with a single subsite of tumors in the gingivobuccal region. All patients were uniformly treated. Following surgical resection of tumors, patients received adjuvant radiotherapy/chemotherapy based on pathological parameters (pTNM staging) in accordance with NCCN guidelines [36]. The immune cells were phenotyped on resected surgical specimens at the center of tumor and invasive margins. We had a long follow-up period for our patients with a median of 50 months for our survival analyses. To the best of our knowledge, there are no studies that investigated TILs exclusively in GBOSCC. Based on our study, we propose that quantification of density of CD3, and CD8 cells in primary tumors may predict disease progression in GBOSCC and could be potentially incorporated into routine histopathological diagnostic assessment. Rapid immune-pathological methods to predict metastasis, may also expedite the management and treatment of the disease. Characterization of the immune contexture will help identify targets for immunotherapy in OSCC-GB. A prospective study is in progress to understand the relationship between genomic alterations and tumor immune microenvironment in GBOSCC employing multi-parameter flow cytometry with simultaneous IHC based topographic assessment of immune cells. Investigating this homogeneous cohort could further provide more insight into the complex tumor immune microenvironment in OSCC-GB.

## Supporting information

**S1 Fig. Microphotograph (20X) of Immunohistochemical staining of p16 for Human Papillomavirus.** Representative sections show positive, equivocal and negative staining for p16.
(TIF)

**S2 Fig. Microphotograph (20X) of Immunohistochemical staining of Immune markers: Serial sections were stained for immune cell markers.** Red arrows indicate the expression of the markers.
(TIF)

**S3 Fig. Comparison of the densities of immune cells in the center of tumor (CT) and invasive margin (IM)] between Group T1 & T2 and Group 2 T3 & T4.** PerkinElmer inForm software was used to enumerate densities of immune cells. Data represented are median with 95% CI. Open circle denotes T1& T2 and the closed circle denotes T3&T4. Two-tailed Mann Whitney test was performed to test statistical significance. T1& T2 (n = 41) and T3&T4 (n = 53).
(TIF)

**S4 Fig. Comparison of the densities of immune markers in center of tumor (CT) and invasive margin (IM) in primary tumors between lymph node positive (N$^+$) and lymph node negative (N0) patients.** PerkinElmer inForm software was used to enumerate densities of immune cells. Data represented are median with 95% CI. Closed circle denotes N$^+$ and the open circle denotes N0. Two-tailed Mann Whitney test was performed to test statistical

significance. $N^+$ (n = 52) and N0(n = 42).
(TIF)

**S1 Table.**
(DOCX)

## Acknowledgments

We thank the Dr. Kent Jensen, Stanford University School of Medicine, for critically reading the manuscript. We thank Indian Statistical Institute, Kolkata, India for assisting us in depositing patient data to ISI SyMeC Cancer Database (ISI-SyMeC-CDb).

## Author Contributions

**Conceptualization:** Geetashree Mukherjee, Partha Pratim Majumder, Suparna Dutt.

**Data curation:** Swarnendu Bag.

**Formal analysis:** Swarnendu Bag, Partha Pratim Majumder.

**Funding acquisition:** Partha Pratim Majumder.

**Investigation:** Geetashree Mukherjee.

**Methodology:** Geetashree Mukherjee, Prasenjit Chakraborty.

**Project administration:** Samrat Roy, Prateek Jain.

**Supervision:** Geetashree Mukherjee, Partha Pratim Majumder, Suparna Dutt.

**Validation:** Geetashree Mukherjee, Debdeep Dey, Paromita Roy.

**Writing – original draft:** Geetashree Mukherjee.

**Writing – review & editing:** Richie Soong, Partha Pratim Majumder, Suparna Dutt.

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
