## [Decision Letter · Decision Letter 0]

24 Jul 2020

PONE-D-20-17762

Association of densities of CD3+ and CD8+ cells in tumors with lymph node metastasis and survival in Gingivo-buccal Oral Squamous Cell Carcinoma

PLOS ONE

Dear Dr. Dutt,

Thank you for submitting your manuscript to PLOS ONE. After careful consideration, we feel that it has merit but does not fully meet PLOS ONE’s publication criteria as it currently stands. Therefore, we invite you to submit a revised version of the manuscript that addresses the points raised during the review process.

We look forward to receiving your revised manuscript.

Kind regards,

Noel F. C. C. de Miranda

Academic Editor

PLOS ONE

Journal Requirements:

2.Thank you for including your ethics statement:

'This study was approved by the Institutional Review Board of TMC (IRB No:EC/TMC/69/16)'.

(a) Please amend your current ethics statement to include the full name of the ethics committee/institutional review board(s) that approved your specific study.  

(b) Once you have amended this/these statement(s) in the Methods section of the manuscript, please add the same text to the “Ethics Statement” field of the submission form (via “Edit Submission”).

3. In the ethics statement in the manuscript and in the online submission form, please provide additional information about the patient records used in your retrospective study, including: a) whether all data were fully anonymized before you accessed them; b) the date range (month and year) during which patients' medical records were accessed; c) the date range (month and year) during which patients whose medical records were selected for this study sought treatment. If patients provided informed written consent to have data from their medical records used in research, please include this information.

4. At this time, we ask that you please provide scale bars on the microscopy images presented in Figure 1 and refer to the scale bar in the corresponding Figure legend.

5. To comply with PLOS ONE submission guidelines, in your Methods section, please provide additional information regarding your statistical analyses. For more information on PLOS ONE's expectations for statistical reporting, please see https://journals.plos.org/plosone/s/submission-guidelines.#loc-statistical-reporting.

6.Thank you for stating the following in the Acknowledgments Section of your manuscript:

[copy in statementWe thank the Department of Biotechnology, Government of India, for funding the project under

Systems Medicine Cluster (Project name: “Multi-dimensional Research to Enable Systems

Medicine: Acceleration Using a Cluster Approach”. Reference No: BT/Med-

II/NIBMG/SyMeC/2014/Vol. II, dated 09/01/2017).]

 [The funders had no role in study design, data collection and analysis, decision to publish, or preparation of the manuscript.]

Reviewers' comments:

Reviewer's Responses to Questions

**Comments to the Author**

1. Is the manuscript technically sound, and do the data support the conclusions?

Reviewer #1: Yes

Reviewer #2: Yes

2. Has the statistical analysis been performed appropriately and rigorously? 

Reviewer #1: Yes

Reviewer #2: No

3. Have the authors made all data underlying the findings in their manuscript fully available?

Reviewer #1: Yes

Reviewer #2: Yes

4. Is the manuscript presented in an intelligible fashion and written in standard English?

Reviewer #1: No

Reviewer #2: Yes

5. Review Comments to the Author

Reviewer #1: The manuscript «Association of densities of CD3+ and CD8+ cells in tumors with lymph node metastasis and survival in Gingivo-buccal Oral Squamous Cell Carcinoma» explores the abundance and distribution of various immune cells in a homogenous cohort of gingivo-buccal OSCC and it’s association with size, lymph node metastases and survival. The analyses are mostly well performed and well described, but some sentences needs revision as specified below. The homogenous cohort is an important strength of the study.

Comments

1. The title is intricate and should be rephrased (Density of CD3+ and CD8+ cells in gingiva-buccal oral squamous cell carcinoma is associated with lymph node metastases and survival).

2. Tumor center is more logically be abbreviated TC

3. Abstract need linguistic revision. Also, the statement. “Furthermore, high densities of CD3+ and CD8+ cells in CT resulted in a significantly longer patient survival” should be revised as the term “resulted in” suggests a causal relationship that cannot be determined based on this type of study.

4. Introduction: This was first described by Galon et al. in colorectal cancer [17, 19], where the density of T cells was measured as a “score” both at the invasive tumor margin and tumor center. Subsequently, the consensus immunoscore categorizing inflamed and non-inflamed tumors was validated globally with high clinical relevance [25]. Reference 25 only includes colorectal cancer, thus this statement should be stated as globally can be misinterpreted as to be valid for all types of cancer.

5. Materials and Methods:

a. Patient tumor tissue: had any of the included patients had prior treatment at TMC? What version of the TNM classification was used? Is the N-status clinically or pathologically determined? Please include references to guidelines.

b. Slide image analyses: “For each slide, 5 different regions of tumor, including the tumor center and margins, were captured.” Which regions, except tumor center and margins were captured? How many images were assessed for each region? “ Immune cell levels were quantified as the percentage of cells with a minimal intensity that was considered positive [35].” It is difficult to grasp this description of how the staining was quantified, please add a more thorough description. How was the reproducibility of scoring between the two observers? It would be informative to include the definitions of immune hot and immune altered phenotypes.

6. Results:

a. Patient characteristics: These numbers could have been displayed in a table.

7. Discussion:

a. “One study has shown that the risk of malignant transformation from leukoplakia is high in oral cancers when the total CD3+cells in the TME are less in contrast to the presence of high density of CD3+ in leukoplakia that did not transform to squamous cell carcinoma [39]” Sentence need linguistic revision.

b. “Our results are in concordance with existing reports [40-42] that pre-existing tumor infiltrating CD8+T cells are strongly associated with improved patient survival.” What is meant by pre-exiciting?

c. “Another study used tissue microarrays (TMA) [43]” There is no information about what was found in this study.

d. “There are no studies with specific or homogeneous sub sites.” This is not true, at least this paper assesses immune infiltrate in OSCC, on whole tissue sections: DOI: 10.1038/s41379-018-0019-5

e.

8. General: Multivariate analyses would strengthen the study.

9. For readers not familiar with immune cell markers it would be informative if the type of immune cells recognized by the various markers were specified.

Reviewer #2: See attachment as well.

Overall impression:

Mukherjee et al investigated the relationship between T and N stage and the expression of 12 markers (CD3, CD8, CD4, Granzyme B, CD68, neutrophil elastase, HLADR, CD15, CD14, arginase1, CD56 and CD20) in the invasive margin (IM) and tumor center (CT) in a cohort of 94 gingiva-buccal OSCC patients. They found that higher densities of CD3+ cells were associated with lower T stage and, and higher densities of both CD3+ and CD8+ cells were associated with the absence of lymph node metastases and better survival.

They performed an elaborate study of 12 immune related markers, in a gingiva-buccal OSCC dedicated cohort. The research question is straightforward, the findings are of interest, however there are some points that need to be addressed.

Major issues:

- It would be essential to know the HPV status of the tumors, as HPV infection is known to result in immunologically hot tumors, better response to therapy and improved survival. Hence, if the HPV status of the tumors is unknown, performing e.g. an additional P16 staining would be very important, to rule out that that is a potential confounder in this study.

- Why was the relationship of these 12 markers with M stage not investigated, but only with T and N stage?

- Regarding the statistical analyses: was the data normally distributed, i.e. was a test for normality performed? If not, that would be necessary. And if the data is not normally distributed, medians instead of means should be presented in the text and figures, and e.g. Mann-Whitney tests should be performed instead of T-tests.

- How do the authors biologically explain the differences in findings between IM and CT? (e.g. improved survival was only observed in patients with high densities of CD3+ and CD8+ cells in the CT, but no such association was seen in the IM)

- Paragraph ‘Association of Tumor Immunoscore with Lymph Node Metastasis’: why was this association of Tumor Immunoscore not also investigated with T and M stage as well?

- Results section, sentence ‘The association of markers’ expression (CT) with node status was further evaluated through ‘line of best fit’ method.’ Why only CT, and not IM as well?

Minor issues:

Introduction:

- ‘The morbidity and mortality associated with OSCC-GB remains poor’ quantify poor

- ‘The consensus immunoscore’ only in colorectal cancer it has recently reached consensus, in most other cancer types not yet, this needs rephrasing.

- ‘Grading and staging were performed according to the WHO classification of tumors and UICC TNM classification’ missing the references here

- ‘All patients received standard-of-care treatment according to National Comprehensive Cancer Network (NCCN) guidelines’ missing references

Methods:

- ‘After washing, peroxidase blocking was carried out’ with what were the washing steps performed?

- The rationale behind the selection of these 12 markers is lacking, please add this.

- Were consecutive slides used for the 12 marker single IHC stainings?

- ‘at 10x and 20x magnification’ it is not clear what the authors mean by this. Were images of all tumors taken at both 10x and 20x magnification? Which images were used for the analyses?

- ‘For each slide, 5 different regions of tumor, including the tumor center and margins, were captured’ 5 regions of both CT and IM together (and if so, how many of which), or 5 regions of CT and 5 regions of IM? And how were the regions selected? This needs to be elaborated.

- ‘The final score was an average from independent assessments of histopathological images by of two onco-pathologists.’ Was the score obtained using InForm software used, or the scoring performed by the pathologists? It is unclear which data was obtained using InForm, and which data from scoring by the pathologists, also in the results section.

- ‘Each patient was re-contacted and relevant data collected for at least two years by telephone calls or during their follow up visits to the hospital.’ What was the relevant data that was collected?

Statistical analysis:

- ‘explanatory variable’ such as?

- ‘Multiple testing correction was done, wherever appropriate’ how was this correction performed? And when was this considered ‘appropriate’? Please elaborate.

- ‘K-Nearest Neighbors (KNN) based supervised machine learning algorithm was used to predict any plausible classification between the expression of immune cell markers and node positivity.’ Why not also between the expression of immune cell markers and T stage?

Results:

- Be consistent in fully writing out the numbers or not. Here, not writing them full out would be clearer (e.g. ‘Ninety-five percent’ as 95%).

- The 95% confidence intervals are missing behind each p-value, please add these.

- ‘No statistically significant difference (p>0.05) in the expression of Granzyme B, CD68, neutrophil elastase, CD15, CD14, Arginase1, CD56 were found between the T1&T2 and T3&T4 groups in both the CT and IM (S2Fig).’ add the underlined words. Remove ‘CD68’ in this sentence, as in supplemental figure 2 and in the following sentence it becomes apparent that there are differences in CD68 expression between the two groups.

Discussion:

- ‘HLA-DR is recognized as a marker of T cell activation [48]’ that is not correct, HLA-DR is required for T cell activation, but is not in itself a marker of T cell activation.

- ‘All patients were uniformly treated’ how is that? Please specify

- ‘The results in different studies are variable because of the non-specific nature of staining. Therefore, CD68 cannot be considered a marker of significance.’ The statement that CD68 cannot be considered a marker of significance is not supported by many recent studies, that have found that high CD68 expression is frequently associated with better survival and response to (immune)therapies.

- ‘Although we found high densities CD4+ cells in the tumors associated with lymph negative disease, this marker is expressed on T cells as well as macrophages and monocytes [47]. Therefore, staining for CD4 expression alone does not distinguish between these cells and CD4+ T cells.’ What is the relevance of these sentences related to this study? They currently seem abundant, and would better be removed.

Figures:

- Figure 4A: unclear what the axes mean, what are the units of the X-axes and Y-axes? This figure needs a more elaborate figure legend.

- Figure 5: write in each Kaplan-Meier graphs behind ‘High’ and ‘Low’ ‘n=……’ for each graph line respectively

Other comments:

There are quite some double spaces and interpunction errors (e.g. dot instead of comma in an enumeration). Please thoroughly go through the manuscript to remove these errors.

Some additional formulation mistakes:

Page 3: ‘The composition of the tumor immune microenvironment’

Page 4: ‘such as chromosomal instability and mutations’ replace mutations by mutation burden

Page 4: ‘Immuno-histochemical’ remove dash, consistently without dash

Page 4: ‘clinical and pathology details of each patient were collected from hospital records’

Page 6: ‘assessments of histopathological images by of two onco-pathologists.’ Remove of

Page 8: ‘Patients with high percentages of CD3+, CD8+ and CD68 + cells at the tumor centers and invasive margins were more frequently lymph node metastasis negative.’

Page 8: ‘while a high HLA-DR expression at the invasive edge was associated with lymph node metastasis.’ Remove ‘a’

Page 9: ‘Fig 4B showed that ‘High’ and ‘Intermediate’ immunoscore are significantly (p<0.05) increased in patients with node negative status compare with node positive patients.’

Page 10: ‘However, there are no studies where CD3+ cell count has been correlated with tumor size.’

Page 11: ‘The lack of T cell infiltration in the tumor core could indicate immunological ignorance - inability of adaptive immunity to recognize tumors [46].’

Page 11: ‘In our study high CD68 density had association with lymph node negative disease.’

Figures 1, 2, 3: make these figures in portrait, not landscape, orientation

Supplemental figure 1: write the name of the marker in the image, instead of the numbers, that makes it clearer to the reader.

Thank you.

6. PLOS authors have the option to publish the peer review history of their article (what does this mean?). If published, this will include your full peer review and any attached files.

Reviewer #1: No

Reviewer #2: **Yes: **Ziena Abdulrahman

---

## [Author Response · Author response to Decision Letter 0]

18 Sep 2020

The response to reviewers is provided in a separate file.

---

## [Decision Letter · Decision Letter 1]

27 Oct 2020

Density of CD3+ and CD8+ cells in Gingivo-Buccal Oral Squamous Cell Carcinoma is Associated with Lymph Node Metastases and Survival

PONE-D-20-17762R1

Dear Dr. Dutt,

We’re pleased to inform you that your manuscript has been judged scientifically suitable for publication and will be formally accepted for publication once it meets all outstanding technical requirements.

Kind regards,

Noel F. C. C. de Miranda

Academic Editor

PLOS ONE

Additional Editor Comments (optional):

Please do address the additional comment by Reviewer 2 in the final version of your manuscript.

Reviewers' comments:

Reviewer's Responses to Questions

**Comments to the Author**

1. If the authors have adequately addressed your comments raised in a previous round of review and you feel that this manuscript is now acceptable for publication, you may indicate that here to bypass the “Comments to the Author” section, enter your conflict of interest statement in the “Confidential to Editor” section, and submit your "Accept" recommendation.

Reviewer #1: All comments have been addressed

Reviewer #2: All comments have been addressed

2. Is the manuscript technically sound, and do the data support the conclusions?

Reviewer #1: Yes

Reviewer #2: Yes

3. Has the statistical analysis been performed appropriately and rigorously? 

Reviewer #1: Yes

Reviewer #2: Yes

4. Have the authors made all data underlying the findings in their manuscript fully available?

Reviewer #1: Yes

Reviewer #2: Yes

5. Is the manuscript presented in an intelligible fashion and written in standard English?

Reviewer #1: Yes

Reviewer #2: Yes

6. Review Comments to the Author

Reviewer #1: Comments have been adequatly addressed. I have no further comments. .................................

Reviewer #2: The authors have done a nice job incorporating the comments.

It is interesting to see that only 4 out of 94 tumors were (equivocal) HPV positive (using P16 IHC as a surrogate marker for HPV), in order to put this into perspective, it would be relevant if the authors could add the percentage of HPV positive tumors as generally found in literature for OSCC.

7. PLOS authors have the option to publish the peer review history of their article (what does this mean?). If published, this will include your full peer review and any attached files.

Reviewer #1: No

Reviewer #2: **Yes: **Ziena Abdulrahman

---

## [Editor Report · Acceptance letter]

9 Nov 2020

PONE-D-20-17762R1 

Density of CD3^+^ and CD8^+^ cells in Gingivo-Buccal Oral Squamous Cell Carcinoma is Associated with Lymph Node Metastases and Survival 

Dear Dr. Dutt:

I'm pleased to inform you that your manuscript has been deemed suitable for publication in PLOS ONE. Congratulations! Your manuscript is now with our production department. 

Kind regards, 

on behalf of

Dr. Noel F. C. C. de Miranda 

Academic Editor

PLOS ONE